# The Role of the Oral Microbiota in the Etiopathogenesis of Oral Squamous Cell Carcinoma

**DOI:** 10.3390/microorganisms9081549

**Published:** 2021-07-21

**Authors:** Tereza Vyhnalova, Zdenek Danek, Daniela Gachova, Petra Borilova Linhartova

**Affiliations:** 1Environmental Genomics Research Group, RECETOX, Faculty of Science, Masaryk University, Kamenice 5, 62500 Brno, Czech Republic; 493682@mail.muni.cz (T.V.); 460558@mail.muni.cz (D.G.); peta.linhartova@gmail.com (P.B.L.); 2Department of Maxillofacial Surgery, Faculty of Medicine, Masaryk University, Jihlavská 20, 62500 Brno, Czech Republic; 3Department of Maxillofacial Surgery, University Hospital Brno, Jihlavská 20, 62500 Brno, Czech Republic; 4Institute of Medical Genetics and Genomics, Faculty of Medicine, Masaryk University, Kamenice 5, 62500 Brno, Czech Republic

**Keywords:** oral microbiome, *Porphyromonas gingivalis*, *Candida* sp., oral squamous cell carcinoma, oral cancer, oral carcinogenesis, tumor microenvironment

## Abstract

Dysbiosis in the oral environment may play a role in the etiopathogenesis of oral squamous cell carcinoma (OSCC). This review aims to summarize the current knowledge about the association of oral microbiota with OSCC and to describe possible etiopathogenetic mechanisms involved in processes of OSCC development and progression. Association studies included in this review were designed as case–control/case studies, analyzing the bacteriome, mycobiome, and virome from saliva, oral rinses, oral mucosal swabs, or oral mucosal tissue samples (deep and superficial) and comparing the results in healthy individuals to those with OSCC and/or with premalignant lesions. Changes in relative abundances of specific bacteria (e.g., *Porphyromonas gingivalis*, *Fusobacterium nucleatum*, *Streptococcus* sp.) and fungi (especially *Candida* sp.) were associated with OSCC. Viruses can also play a role; while the results of studies investigating the role of human papillomavirus in OSCC development are controversial, Epstein–Barr virus was positively correlated with OSCC. The oral microbiota has been linked to tumorigenesis through a variety of mechanisms, including the stimulation of cell proliferation, tumor invasiveness, angiogenesis, inhibition of cell apoptosis, induction of chronic inflammation, or production of oncometabolites. We also advocate for the necessity of performing a complex analysis of the microbiome in further studies and of standardizing the sampling procedures by establishing guidelines to support future meta-analyses.

## 1. Introduction

Oral squamous cell carcinoma (OSCC) is the most common malignancy in the head and neck region. In 2020, the number of new cases of OSCC worldwide was 377,713, and the number of deaths in the same year was 177,757 [1]. Patients with this diagnosis have a low survival rate and poor prognosis. Despite advances in the surgical and systemic treatment of this disease, the 5 year survival for patients diagnosed with later stages of the disease remains below 50% [2].

Oral carcinogenesis is a multifactorial process involving the effect of the exposome and subsequent cytogenetic and epigenetic changes in keratinocytes. Components of the external exposome, such as alcohol and tobacco consumption, poor oral hygiene, or poor dietary habits, are generally considered to be the dominant etiological factors of this disease [3]. At present, the effects of oral microbiota, i.e., internal exposome, are intensively studied in association with the initiation and progression of OSCC.

Chronic inflammation was shown to be closely related to carcinogenesis. Microbiota can participate in both of these processes, for example, through modulating the immune response of the host [4]. Substances such as reactive oxygen species, reactive nitrogen intermediates, or cytokines produced by immune cells can contribute toward the initiation of carcinogenesis through induction of mutations, genomic instability, or epigenetic changes. Proinflammatory cytokines subsequently activate key transcription factors such as STAT3 (signal transducer and activator of transcription) or NF-κB (nuclear factor kappa B) in premalignant lesions, and these mechanisms promote other malignant processes, such as cell proliferation, angiogenesis, or metastasis, and maintain the inflammatory tumor microenvironment. In addition to inducing chronic inflammation, the oral microbiota has also been linked to OSCC through the production of oncometabolites, induction of epithelial–mesenchymal transition (EMT), inhibition of apoptosis, or stimulation of cell proliferation [5,6].

Bacteria are the largest contributors to the composition of the oral microbiota, with fungi and viruses constituting a smaller proportion. A healthy microbial community is stable; however, under certain conditions, microbial homeostasis can be disrupted and a state of dysbiosis develops, which is characterized by an increased representation of microorganisms with pathogenic potential or increased expression of virulent factors. These microorganisms can reverse the relationship with the host from mutualistic to parasitic [2].

Microorganisms colonize various parts of the oral cavity including teeth, gingival sulcus, tongue, buccal mucosa, hard and soft palate, or tonsils; the composition of the microbiota differs with sites [7]. Nejmen et al. reported that the microbiota within tumors can reside intracellularly in both the tumor and the immune cells and that the bacteriome varies with the tumor type and subtype, smoking status, and immunotherapy response [8]. Thus, the existence of tumor-related microbiota in a tumor microenvironment may be responsible for driving OSCC progression [9].

The aim of this review was to summarize the current knowledge about the association of bacteria, fungi, and viruses with OSCC and to describe possible etiopathogenetic mechanisms involved in processes of OSCC development and progression.

## 2. Oral Bacteria Associated with Oral Cancer

Many species of anaerobic bacteria have been proposed to be involved in carcinogenesis [10]. Nagy et al. detected significantly higher amounts of *Porphyromonas* and *Fusobacterium* in OSCC tissue samples compared to samples from healthy mucosa [11]. In addition, Katz et al. reported the relative abundance of *Porphyromonas gingivalis* in gingival squamous cell carcinoma to be higher than in healthy gingival tissue samples [12]. Similarly, *Prevotella* spp. were detected in OSCC [13,14]. These bacteria and mechanisms via which they can contribute toward carcinogenesis are discussed below in more detail.

*Filifactor* bacteria were reported to induce the secretion of proinflammatory cytokines, activate specific oncogenes, or maintain the inflammatory state. Interestingly, they also play a role in enhancing tumor progression by promoting colonization of the tumor bed by other pathogens [9]. Similarly, *Tanarella* can also support the production of pro-inflammatory cytokines or secretion of cysteine-like proteases, arresting the cell cycle in the G2 phase [4]. Increased amounts of *Parvimonas* were detected in OSCC. They are known to induce inflammation, but other mechanisms of their association with OSCC have not yet been revealed [15,16]. The genus *Aggregatibacter* is characterized by inducing the production of proinflammatory cytokines, hydrogen sulfide, and methyl mercaptan, which may result in inflammation, cell proliferation, and migration, or tumor angiogenesis [4,9].

*Eikenella corrodens* is also capable of inducing the expression of proinflammatory cytokines, namely, interleukins IL-1, IL-6, and IL-8, and tumor necrosis factor α (TNF-α) [2]. Production of IL-6 and IL-8 can also be elevated by *Propionibacteria* [4]. Cytokine IL-23 overexpression can be induced by *Gemella* [17]. Another bacterium that may be involved in the process of tumor progression, *Treponema denticola*, supports the overexpression of dentilisin, which was associated with increased tumor invasiveness [18].

Aerobic bacteria also constitute an important component of the oral microbiome. Most aerobic bacteria are located in the superficial regions of the oral cavity and act as commensal microorganisms maintaining microbial balance. Only a few species of aerobic bacteria have been reported to have pathogenic properties [10].

Aerobic bacteria that have been associated with OSCC include *Streptococcus* spp. and *Lactobacillus* spp., described in detail below. Overrepresentation of *Pseudomonas aeruginosa* in the OSCC microbiome has also been reported [19]; these bacteria were shown to induce inflammation through activation of the NF-κB signaling pathway [9]. Bacterial endotoxins (such as lipopolysaccharides, LPS) or structural parts (such as flagella) can also be associated with the inflammatory state in the organism [16]. *Pseudomonas aeruginosa* can, through its enzyme c1 nitrite reductase, reduce nitrites to nitric oxide [3]. It is also capable of secretion of the LasI factor, leading to the downregulation of a tumor-suppression protein E-cadherin.

*Rothia* are significant producers of the carcinogenic acetaldehyde; increased representation of these bacteria has been detected in OSCC [9]. Other bacteria overrepresented in OSCC include, for example, *Mycoplasma salivarium*, as well as genera *Parvimonas* and *Dialister* [10,20]. Contrary, *Streptomyces* have been reported to have a tumor-protective effect thanks to their capability to induce apoptosis of tumor cells [21].

Mechanisms of association of selected aerobic and anaerobic bacteria with tumors of the oral cavity are presented in Table 1 [4,9,15,16,17,18,21,22,23,24,25,26,27,28,29,30,31,32,33,34,35,36,37,38]; an overview of selected bacteria most frequently identified in association with OSCC is provided in Appendix A [12,13,14,39,40,41,42,43,44,45,46,47,48].

### 2.1. The Role of Porphyromonas gingivalis in Oral Cancer

*P. gingivalis* is considered one of the key periodontal bacteria with pathogenic potential that can be found in the oral cavity. It has been shown to colonize malignancies such as OSCC, esophageal squamous cell carcinoma, and gingival carcinoma [49]. A meta-analysis by Sayehmiri et al. reported that the presence of *P. gingivalis* increases the risk of developing OSCC more than 1.36-fold [50]. The mechanisms via which this Gram-negative bacterium interferes with tissue integrity and disrupts the host immune response include inhibition of cell apoptosis, activation of cell proliferation, induction of chronic inflammation, and production of oncometabolites [51].

#### 2.1.1. Inhibition of Cell Apoptosis

Epithelial cells represent the initial point of entry of microorganisms into the host organism and, thus, the first line of defense against microbial pathogens. Elevated levels of bacteria, including *P. gingivalis*, have been shown to modulate apoptotic pathways in these cells [52]. Apoptosis was inhibited in epithelial cells highly infected by *P. gingivalis*, thus manifesting an antiapoptotic phenotype strongly associated with carcinogenesis [53].

*P**. gingivalis* can induce inhibition of apoptosis through overstimulation of the JAK1/STAT3 signaling pathway, participating in the regulation of mitochondrial apoptosis, cellular differentiation, migration, and proliferation [54]. *P. gingivalis* has a lipopolysaccharide complex, which can bind to the TLR-4 receptor on the surface of human cells, thus activating the NF-κB protein complex that subsequently enters the nucleus and activates the transcription of cytokine genes, such as the gene for interleukin 6 (IL-6). IL-6 can then serve as a ligand for a receptor associated with JAK1 kinase and, thus, facilitate constitutive activation of the STAT3 transcription factor, which leads to the overexpression of antiapoptotic genes, thus inhibiting apoptosis and contributing to the proliferative phenotype of the cell [31].

Another mechanism via which *P. gingivalis* can participate in the inhibition of cell apoptosis is the upregulation of miRNA-203. MicroRNA (miRNA) is a small noncoding RNA molecule characterized by the ability to bind to the 3′ UTR (untranslated region) of the target mRNA, subsequently degrading this mRNA and thereby suppressing the expression of the particular gene. These molecules are involved not only in the regulation of apoptosis but also in cell differentiation and host immune response [55]. miRNA-203 is known to target the 3′ UTR region of the suppressor of cytokine signaling 3 (SOCS3), thus downregulating its expression. SOCS3 belongs to the SOCS family of proteins that provide negative regulation of the JAK/STAT signaling pathway by inhibiting the enzymatic activity of Janus kinase [56]. It has been found that gingival epithelial cells that are infected with *P. gingivalis* show increased levels of miRNA-203, whereas SOCS3 levels are decreased. The mechanism via which *P. gingivalis* upregulates miRNA-203 is not yet fully understood, but some authors believe that this occurs through activation of the activator protein AP-1 by protein kinase C [57]. Thus, inhibition of SOCS3 by *P. gingivalis* is another mechanism leading to the constitutive activation of the JAK1/STAT3 signaling pathway, thereby suppressing host cell apoptosis.

*P. gingivalis* can, however, also induce proapoptotic effects through the production of nucleoside diphosphate kinases. Nucleoside diphosphate kinase (NDk) is one of the ubiquitously occurring enzymes across the eukaryotic and prokaryotic domains. It has an indispensable role in nucleotide metabolism, catalyzing the transfer of phosphate from NTP (nucleoside triphosphate) to NDP (nucleoside diphosphate). Nucleoside diphosphate kinases, which are secreted by *P. gingivalis*, may, among other functions, work as effector proteins in the regulation of cellular apoptosis [58]. As shown in several studies, bacterial NDks can induce apoptosis through phosphorylation of serine residues of the heat-shock protein 27 (Hsp27) [59]. These proteins can interact with key components of the apoptotic pathway, thereby regulating it. When Hsp27 proteins are phosphorylated, their oligomerization and subsequent inactivation of the proapoptotic protein Bax occur, preventing cytochrome c efflux from the mitochondrial intermembrane space and ultimately inhibiting apoptosis [60].

#### 2.1.2. Activation of Cell Proliferation

Proteomic studies have shown that infection of gingival epithelial cells (GECs) with *P. gingivalis* results in changes in the concentrations of certain proteins and their phosphorylation, which interferes with cell-cycle regulation. Specifically, pathways involving cyclins, cyclin-dependent kinases, and p53 protein are affected by these bacteria [61,62].

*P. gingivalis* accelerates the progression of epithelial cells through the S and G2 phases of the cell cycle via upregulation of cyclins, subsequent activation (phosphorylation) of cyclin-dependent kinases, and downregulation of the tumor suppressor protein p53. Thus, in addition to its ability to inhibit apoptosis, *P. gingivalis* may also contribute to an increase in the cell proliferation through this mechanism.

Infection of epithelial cells with *P. gingivalis* also leads to the increased expression of the regulatory protein beta-catenin. This transcription factor, in addition to upregulation of cyclins, leads to overexpression of ZEB1 (zinc finger E-box-binding homeobox 1). This regulatory protein inhibits the formation of the tumor suppressor protein E-cadherin and induces epithelial–mesenchymal transition, during which cells lose their polarity and transform into motile mesenchymal cells. This process is also associated with tumor progression or metastasis formation [63].

Beta-catenin signaling is an important pathway affecting carcinogenesis and cell proliferation. *P. gingivalis* has been shown to induce noncanonical activation of beta-catenin and proteolytic dissociation of the beta-catenin destruction complex through the production of the gingipain family proteases [62]. The manner in which *P. gingivalis* affects the beta-catenin signaling pathway may represent a novel mechanism of proteolytic action via which the bacterium contributes to the disruption of the oral tissue homeostasis and subsequently to the proliferative phenotype [64].

#### 2.1.3. Induction of Chronic Inflammation

Induction of chronic inflammation is considered to be another potential pathway via which *P. gingivalis* participates in oral carcinogenesis. Chronic inflammation is known to contribute significantly to the OSCC growth, mainly by modulating its microenvironment with cytokines and chemokines [18]. OSCC cells infected with *P. gingivalis* have increased secretion of cytokines, such as IL-8, IL-6, TGF-β1 (transforming growth factor β1), and TNF-α. These polypeptide mediators produced by epithelial cells then become part of the microenvironment of the tumor (e.g., OSCC) and contribute to its progression.

The proinflammatory chemokine IL-8 may be involved in the upregulation of zinc-dependent proteins, so-called matrix metalloproteinases, which facilitate metastasis of malignant epithelial cells by degrading the extracellular matrix. It can also stimulate cell proliferation through transactivation of the epidermal growth factor (EGF) [18].

TGF-β1 is a homodimer protein with the ability to regulate many cellular processes. According to several studies, TGF-β1 is associated with the induction of epithelial–mesenchymal transition, tumor angiogenesis, or metastasis. Similar to IL-8, TGF-β1 may increase the invasiveness of OSCC cells through the activation of matrix metalloproteinases (in particular, MMP-2) [65].

TNF-α is one of the cytokines capable of influencing carcinogenesis at several stages. The protein has been associated with the generation of reactive oxygen species (ROS) or reactive nitrogen intermediates (RNI), which in turn contribute to genomic instability or mutations that are strongly associated with cancer development. Other mechanisms of its action include induction of epithelial–mesenchymal transition, as well as secretion of vascular endothelial growth factor (VEGF), stimulating tumor angiogenesis [66].

*P. gingivalis*, upon entering an epithelial cell, intracellularly secretes the enzyme serine phosphatase (SerB), which dephosphorylates the serine residue of the transcription factor NF-κB. This subsequently inhibits NF-κB translocation to the nucleus, which leads to suppression of IL-8 production [67]. Although this mechanism contributes to the suppression of tumor progression, it may be offset by the inhibition of the angiostatic cytokines CXCL9, CXCL10, and CXCL11; this can promote, in addition to tumor neovascularization, tumor growth or metastasis formation [18].

*P. gingivalis* also possesses the ability to upregulate B7-H1 and B7-DC receptors. The increased expression of B7-H1 proteins, which has been detected in the vast majority of human carcinomas, leads to anergy, i.e., an insufficient immune system response, and to apoptosis of T-lymphocytes, thus supporting tumor cells in overcoming the body’s antitumor response [68].

#### 2.1.4. Production of Oncometabolites

Metabolites produced by *P. gingivalis*, such as oxygen radicals, butyrate, or acetaldehydes, may also be associated with carcinogenesis. Butyrate is one of the most harmful bacterial carcinogens, capable of inducing apoptosis in immune cells such as T-lymphocytes and B-lymphocytes. In addition, it can also trigger the production of oxygen radicals that may cause DNA double-strand breaks or nucleic acid base modifications [34].

Another product of microbial metabolism with proven carcinogenic effects, acetaldehyde, is also produced by *P. gingivalis* by ethanol metabolization. It can be produced in concentrations capable of damaging DNA, thus inducing mutations and hyperproliferation of epithelial cells [64].

### 2.2. The Role of Fusobacterium nucleatum in Oral Cancer

*F. nucleatum* is a Gram-negative anaerobic bacterium forming a natural part of the oral microbiota; in patients with head and neck cancer, however, it can be detected at elevated levels. *F. nucleatum* may play an essential role in the development of oral cancer and has been associated with tumorigenesis by several mechanisms [18].

#### 2.2.1. Secretion of IL-1β Due to NLRP3 Inflammasome Activation

NLRP3 (NOD-like receptor 3) is a large multiprotein complex belonging to the NLR family of inflammasomes, playing an essential role in the body’s defense against bacterial, fungal, and viral infections. The NLRP3 inflammasome consists of the NLRP3 protein (containing the central nucleotide-binding domain NACHT, the N-terminal pyrin domain PYD, and the C-terminal leucine-rich LRR domain) and the ASC protein (apoptosis-associated speck-like protein containing CARD), which facilitates the incorporation of procaspase 1 into the multiprotein complex [69].

Infection of gingival epithelial cells by *F. nucleatum* leads to an early activation of NF-κB, which subsequently translocates to the nucleus where the pro-interleukin-1β (pro-IL-1β) gene is expressed. The NLRP3 inflammasome is simultaneously activated by bacterial infection. Formation of the NLRP3 inflammasome induces autocatalytic activation of caspase 1, which subsequently cleaves pro-IL-1β and pro-IL-18 into their biologically active proinflammatory forms IL-1β and IL-18 [27,69].

IL-1β is associated with the development of OSCC, and its elevated levels have also been detected in samples of this tumor. It appears that IL-1β is one of the most important proinflammatory cytokines involved in cancer pathogenesis. IL-1β promotes the malignant transformation of oral dysplastic cells by increasing their proliferation. Overexpression of IL-1β leads to enhanced metastasis (one of the major causes of adverse prognosis in patients with advanced OSCC) through secretion of oncogenic cytokines IL-6, IL-8, and GRO-α (growth-regulated oncogene α) by tumor cells [70].

In relation to OSCC, IL-8 exhibits a dual action. On the one hand, it inhibits tumor growth by inducing apoptosis of transformed cells through the activation of glycogen synthase kinase 3 beta (GSK3B); on the other hand, however, it has been shown in vitro to be able to induce epithelial–mesenchymal transition, which plays an essential role in the formation of tumor metastases [71].

It was found that, once caspase 1 is activated in *F. nucleatum*-infected cells, the DNA-associated high mobility group box protein 1 (HMGB1) is redistributed from the nucleus to the cytosol and extracellular space [27]. HMGB1 is one of the so-called “danger” signals (damage-associated molecular pattern molecules, DAMPs) involved in maintaining the inflammatory response of the organism [35]. The adaptor protein ASC is involved in the activation of the inflammasome; at the same time, it is secreted by cells during NLRP3 formation. Once in the extracellular space, ASC can amplify the immune response and act as a “danger” signal. Thus, both ASC and HMGB1 proteins are capable of amplifying the immune response [27].

#### 2.2.2. Metalloproteinase Overexpression Due to p38 Activation

The expression of matrix metalloproteinase-13 (MMP-13) and metalloproteinase 9 (MMP-9) is stimulated in cells infected with *F. nucleatum*. These bacteria also induce activation of protein kinase p38 in infected cells, which leads to increased secretion of MMP-13 and MMP-9. These metalloproteinases play an important role in tissue metabolism and maintenance of homeostasis; in addition, they are involved in processes such as the body’s inflammatory response and metastasis formation. Excessive secretion of MMP-13 and/or MMP-9 contributes to tumor invasiveness [28].

#### 2.2.3. Ku70/p53 Signal Pathway-Dependent DNA Damage

DNA damage is known to significantly contribute to the development and progression of oral cancer. Double-strand breaks (DSBs), which can be repaired by a nonhomologous end joining (NHEJ) mechanism that allows direct joining of broken DNA strands, are among the most severe types of DNA damage.

Nonhomologous end joining is initiated by the Ku protein complex, which attaches to any two free DNA ends, allowing the binding of additional enzymes involved in the repair of broken strands by removing the defective nucleotides. The ends are resynthesized to make the strands complementary. The heterodimeric Ku protein consists of Ku70 and Ku80 subunits. Ku70 has been found to be involved in DDR signaling (DNA damage response) through activation of cell-cycle checkpoints and initiation of apoptosis. Once the Ku complex binds to the free ends of DNA, the tumor suppressor protein p53 is also activated, its expression is upregulated, and DNA repair is initiated.

According to recent findings, excessive proliferation of *F. nucleatum*-infected OSCC cells due to DNA damage may be associated with the Ku70/p53 signaling pathway [54]. Normally, when cellular DNA is damaged, Ku70 is acetylated; the expression of p53 is increased, while the level of Ku70 decreases [72]. If the DNA damage in *F. nucleatum*-infected cells is so severe that the levels of the Ku70 protein are too low to ensure efficient DNA repair, the severely defective DNA is not repaired in a timely manner and abnormal proliferation of OSCC cells can occur. Nevertheless, the detailed mechanism of the interplay between *F. nucleatum* and Ku70 remains unclear [29,73].

#### 2.2.4. Acceleration of the Cell Cycle through Downregulation of p27

The tumor suppressor protein p27, a member of the cyclin-dependent kinase inhibitor (CDK) family, participates in cell-cycle regulation by binding to CDKs and subsequently blocking cell entry into the S phase. Downregulation of p27 has been found to lead to tumor proliferation and to correlate with adverse cancer prognosis.

According to Geng et al., reduced p27 levels were detected in *F. nucleatum*-infected cells; the number of cells in the G1 phase of the cell cycle was reduced and the percentage of cells in the S phase was significantly increased. These results are in accordance with several other studies reporting that downregulation of p27 leads to cell-cycle arrest in the S phase and to increased cell proliferation [29].

#### 2.2.5. Induction of Epithelial–Mesenchymal Transition

EMT is an often reversible process, in which epithelial cells acquire the motility and invasiveness of mesenchymal cells; its inappropriate activation in response to aberrant stimuli has been associated with carcinogenesis [36]. Several transcription factors play a key role in the transition from an epithelial to a mesenchymal phenotype, a common feature of which is the inhibition of E-cadherin expression, responsible for the epithelial nature of the cell. The transcription factors of the SNAI1, SLUG, and ZEB families bind directly to the promoter of the E-cadherin gene and suppress its expression.

Noncoding RNAs are also important regulators of EMT. MIR4435-2HG, one of the newly discovered long noncoding RNAs (lncRNAs), has been associated with lung and gastric cancer. Infection of oral epithelial cells with *F. nucleatum* was found to lead to upregulation of MIR4435-2HG, which can subsequently specifically bind to another noncoding RNA, namely, microRNA-296-5p, thus downregulating its expression. This mechanism then impairs the ability of microRNA-296-5p to silence the expression of its target gene Akt2, which is then able to activate the expression of the transcription factor SNAI1 and contribute to the transition into the mesenchymal phenotype of infected oral epithelial cells [30].

### 2.3. Role of Prevotella *sp.* in Oral Cancer

The genus *Prevotella* includes Gram-negative bacteria associated with the pathogenesis of periodontitis. *Prevotella* spp. have been detected in OSCC [13,14].

*P. intermedia* produces virulence factors such as lipopolysaccharides, peptidoglycans, or lipoteichoic acid, inducing the production of proinflammatory cytokines [9], including the inflammatory interleukins IL-1, IL-6, IL-17, IL-23, and TNF-α [22]. Cytokine production is further supported by proteases secreted by *P. intermedia*. These proteases can act as signaling molecules through the stimulation of protease-activated receptors (PARs). Such an action can affect apoptosis, as well as cell proliferation or inflammation. Proteases can degrade the extracellular matrix of the host, destroy its physical immune barriers, and modulate the host immune response to support tumor onset and progression [15].

Similar to other periodontal pathogens, *P. intermedia* produces hydrogen sulfate and methyl mercaptan, which are co-responsible for oxidative stress and DNA damage to the oral cells. Hydrogen sulfide inhibits the enzyme superoxide dismutase, which prevents the action of oxygen radicals in cells. Methyl mercaptan is involved in the cleavage of type 4 collagen; the products of this cleavage promote tumor angiogenesis and OSCC invasiveness [4].

### 2.4. Role of Streptococcus *sp.* in Oral Cancer

#### 2.4.1. *Streptococcus anginosus*

*S. anginosus* is a bacterium primarily found in the dental plaque and gingiva. It has been associated with oral carcinogenesis as a factor inducing chronic inflammation through the production of proinflammatory cytokines such as IL-1β, IL-6, or TNFα that can, in turn, contribute to the progression of oral carcinoma [74]. Due to its alcohol dehydrogenase activity, *S. anginosus* is also able to metabolize ethanol to acetaldehyde [32]. According to Sasaki et al., *S. anginosus* was detected in 45% of OSCC samples [75]; nevertheless, it should also be noted that recent studies reported the presence of *S. anginosus* with equal or even higher frequency in nontumor tissues than in tumors [2].

#### 2.4.2. *Streptococcus mitis*

Increased representation of *S. mitis* has been found in the saliva of patients with OSCC. Thus, it has been proposed as a possible early tumor marker [18]. On the other hand, Baraniya et al. demonstrated the in vitro ability of *S. mitis* to inhibit OSCC tumor cell proliferation through cytotoxicity mediated by hydrogen peroxide production [37]. Nevertheless, although such a potential protective action could be attributed to *S. mitis*, these bacteria are also one of the major producers of carcinogenic acetaldehyde and, moreover, can induce the production of proinflammatory cytokines [76].

#### 2.4.3. *Streptococcus gordonii*

*S. gordonii* is a Gram-positive commensal bacterium that occurs naturally in the oral cavity, skin, and/or intestinal tract. *S. gordonii* can, nevertheless, also exhibit pathogenic properties and is associated (for example) with the development of periodontitis or infective endocarditis [77]. *S. gordonii* is antagonistic to *P. gingivalis* as it possesses the ability to induce phosphorylation of serine residues of the transcription factor FOXO1 (forkhead box protein 01). This process is mediated through the TAK1–NLK1 signaling pathway and leads to the inhibition of FOXO1 translocation from the nucleus to the cytoplasm and, therefore, to its activation. The active FOXO1 then upregulates the expression of the ZEB2 factor, which leads to the induction of epithelial–mesenchymal transition [18].

### 2.5. Role of Lactobacillus *spp.* in Oral Cancer

Bacteria of the genus *Lactobacillus* have both carcinogenic and anticarcinogenic effects. They can prevent tumor transformation, for example, by inducing apoptosis, increasing the expression of tumor suppressor genes, or by regulating adaptive and innate immune responses. *Lactobacillus rhamnosus* can suppress chronic inflammation associated with carcinogenesis. Similarly, the *Lactobacillus fermentum* strain has anti-inflammatory effects against oral carcinogenesis. *L. fermentum* can induce apoptosis of oral carcinoma cells through upregulation of PTEN (phosphatase and tensin homolog) and MAPK signaling.

On the other hand, *Lactobacillus bacteria* produce lactic acid, which, together with other organic acids, acidifies the tumor microenvironment and contributes to the progression of OSCC [32]. The decrease in pH caused by the production of organic acid then leads to the suppression of the antitumor immune response or to the stimulation of tumor angiogenesis (which is necessary for the survival and spread of tumor cells) [78].

## 3. Fungi Associated with Oral Cancer

The mycobiome, comprising the combined genome of various fungal species, is an essential part of the human microbiome. Although the mycobiota constitutes only a minor fraction of the oral microbiota, the impact of fungi on host health can be quite broad. Dysbiosis of the oral mycobiota can result in minor diseases, as well as life-threatening systemic infections [79].

Yeasts of the genus *Candida* are the most abundantly represented fungi in the oral mycobiota. In addition, the genera *Cladosporium*, *Aureobasidium*, *Aspergillus*, or *Mallasezia* can be found in the oral mycobiome, as can many others [80].

The relationship between mycobial dysbiosis and OSCC has not yet been explored in detail, mainly due to the relatively low prevalence of individual fungal species and the lack of well-characterized reference genomes [81]. Nevertheless, studies have been published that describe the association between mycobiome and tumor transformation and demonstrate that the diversity of individual fungal species comprising the oral mycobiome is reduced in patients with OSCC. The absence of certain species has been generally associated with head and neck cancers. The yeasts of the genus *Malessezia*, the abundance of which was reduced in patients with cancer compared to healthy individuals, can serve as an example of such a relationship. A similar tendency can also be observed in fungi of the genus *Schizophyllum*, which are capable of producing the polysaccharide schizophyllan with anticancer activity.

On the other hand, an increased prevalence of certain fungal species is associated with oral tumor transformation. For example, an increased incidence of the genera *Candida*, *Hannaella*, and *Giberella* was detected in samples of OSCC tumor tissues [82]. The mycobiota associated with OSCC is summarized in Table 2 [82,83,84].

### 3.1. The Role of Candida *spp.* in Oral Cancer

Yeasts of the genus *Candida* belong to commensal microorganisms colonizing the oral cavity. *Candida* are opportunistic pathogens; thus, depending on the microenvironment or the condition of the host immune defense mechanisms, they can transform from harmless commensals into pathogenic microorganisms that may be involved, among others, in oral carcinogenesis [85]. The relationship between oral cancer transformation and *Candida* yeasts is also implied by the elevated levels of *Candida albicans* or *Candida etchellsii* found in OSCC samples. *Candida albicans*, *dubliniensis*, *tropicalis*, *pintolopesii*, and *glabrata* yeasts have also been detected in premalignant lesions of patients with chronic hyperplastic candidiasis [4].

#### *Candida* *albicans*

*C. albicans* can induce carcinogenesis through its proinflammatory action, through induction of Th17 response, or through the production of direct carcinogens [81]. *C. albicans* is capable of forming nitrosamines that act as carcinogens both alone and when combined with other chemical compounds. Their production leads to the activation of specific proto-oncogenes that can further support the formation of a carcinogenic lesion [85].

*C. albicans*, similar to *C. tropicalis*, *C. parapsilosis*, and *C. glabrata*, possesses alcohol dehydrogenase activity, i.e., it is capable of metabolizing ethanol to acetaldehyde. The action of acetaldehyde in the oral cavity leads to the production of DNA–protein adducts. These aberrant molecules interfere with DNA replication, which leads to point mutations and chromosomal aberrations. Moreover, such adducts also negatively affect enzymes involved in DNA repair, which facilitates the activation of proto-oncogenes and disruption of the cell cycle and, in turn, may result in tumor formation. Acetaldehyde can also bind to the antioxidant glutathione and, thus, reduce the removal of DNA-damaging oxygen radicals.

Endothelial cells recognize pathogen-associated molecular patterns (PAMPs) of *C. albicans* using pattern recognition receptors (PRRs). Signaling resulting from the recognition of microbial patterns by the receptors leads to activation of gene expression and synthesis of cytokines, adhesion molecules, or immune receptors responsible for the proinflammatory and antimicrobial response of the organism. Chemotactic cytokines involved in the immune response include CXCL1, CXCL3, and CXCL3, which are closely associated with tumor transformation and angiogenesis [86]. Other proinflammatory cytokines produced as a result of this signaling include TNF-α and IL-18, which are associated with tumor invasiveness, migration, and metastasis [87].

Overexpression of the cytokines IL-6 and IL-8 by oral tumor cells can also be correlated with the action of *C. albicans*. Both these groups of cytokines can stimulate carcinogenesis; the antiapoptotic effect of IL-6 can also be mentioned. The increased production of IL-6 and IL-8 by tumor cells can be explained through glycosylation of proteins, β-glucans, or chitins of the *C. albicans* cell wall by mannosyl residues or by the production of aspartate proteases (SAPs) by the yeasts [81].

Th-17 CD4 lymphocytes play a very important role in the immune response against *C. albicans*. These helper T-lymphocytes produce cytokines IL-17 and IL-23 [86]. IL-17 cytokines activate the NF-κB family of transcription factors and the Wnt signaling pathway, which can ultimately lead to tumor formation [81]. IL-17 can also induce carcinogenesis indirectly through the recruitment of neutrophils to the tumor site; the presence of these leukocytes in malignant tissues was shown to correlate with a worse disease prognosis. IL-23 cytokines induce tumor angiogenesis and tumor growth. It is also important to mention that IL-23 acts antagonistically to the cytokines IL-12 and interferon gamma (IFN-γ), which are essential for the Th1 antitumor immune response [86].

## 4. Viruses Associated with Oral Cancer

*Herpesviridae* and *Papillomaviridae* are the most common families of the viruses found in the oral cavity in healthy patients [88]. Representatives of *Herpesviridae* residing in the oral cavity include human cytomegalovirus (HCMV), type 1 herpes simplex virus (HSV-1), and Epstein–Barr virus (EBV). From the *Papillomaviridae* family, the human papilloma virus (HPV) is the most common [23].

According to Wang et al., several phage species belonging to the *Siphoviridae*, *Myoviridae*, and *Podoviridae* families have also been detected in saliva and dental plaque samples [89]. Bacteriophages can, generally, play a role in shaping the composition of oral bacterial communities [90,91].

Viruses constituting a part of oral microbiota can be detected both in their active and latent form; they are associated with many diseases of the oral cavity [90]. Over the last years, the role of viruses in the development of oral carcinoma has been the focus of many studies [92,93,94,95].

Selected types of HPV (HPV 16, 18, 31, 33, 35, and 39) possess oncogenic potential [3]. A recent meta-analysis by She et al. proved that EBV infection is statistically significantly associated with a higher risk for OSCC; in the tumor tissue, DNA regions such as viral oncogene BamH1W have been detected [96]. Selected viruses that have been found in the oral carcinomas are presented in Table 3 [97,98,99,100,101,102,103,104,105,106].

### 4.1. The Role of Human Papillomavirus in Oral Cancer

Analysis of 4680 samples in a widely cited study by Miller et al. demonstrated an increased frequency of HPV detection in the mucosa of precancerous lesions and OSCC compared to the healthy mucosa [107]. On the other hand, however, other analyses reported that human papillomavirus was detected at a lower frequency in patients with oral carcinoma than in healthy individuals [4]. According to the current findings, oropharyngeal cancers show higher HPV positivity than OSCC [108].

E6 and E7 viral genes of human papillomaviruses were proposed to be strongly associated with HPV-induced carcinogenesis. The E6 protein can bind to and inhibit the tumor suppressor protein p53 via the ubiquitin-ligase E6AP, which results in the possible suppression of cell apoptosis. Furthermore, inactivation of the proapoptotic proteins Bak and Bax has also been associated with the viral E6 gene. The E7 protein is capable of inhibition of the tumor suppressor protein pRb (retinoblastoma protein), playing a role in cell-cycle regulation [109].

The pRb protein can bind to E2F family transcription factors and repress the transcription of specific genes involved in cell-cycle progression. The E7 viral protein can prematurely induce cell entry into the S phase by disrupting the pRb–E2F complex. If pRb is inhibited by the E7 protein, the p16 protein is also overexpressed.

These mechanisms have been described in association with cervical carcinoma; nevertheless, immunohistochemistry methods also proved excessive expression of p16 in OSCC cells. On the other hand, p53 protein expression has been shown not to correlate with HPV-positive squamous cell carcinoma [110].

### 4.2. The Role of Epstein–Barr Virus in Oral Cancer

Epstein–Barr virus (EBV) is a subcellular organism with a suspected oncogenic potential. EBV is associated, for example, with nasopharyngeal carcinoma or, according to recent findings, Hodgkin’s lymphoma.

A meta-analytic study by She et al. reported a positive correlation of EBV with oral squamous cell carcinoma. EBV-associated carcinogenesis can be attributed to a number of viral proteins regulating cell proliferation and apoptosis. The presence of the viral oncogene *LMP-1* may lead to constitutive activation of NF-κB and inhibition of apoptosis. Other EBV genes include *EBER*s, *EBNA1*, *LMP*-*2*, and *BARF0*; the products of these genes affect both cellular immortalization and viral genome replication. According to She et al., EBV proteins were expressed in most OSCC cells. Nevertheless, although the presence of viral proteins, mRNA, and DNA in OSCC samples strongly implies the existence of a link of EBV to oral carcinogenesis, it is not possible to say with certainty whether or not these mechanisms really contribute to this process [96].

### 4.3. The Role of Human Cytomegalovirus in Oral Cancer

Although HCMV is not generally considered to be an oncogenic virus, several clinical and experimental studies have suggested that cytomegalovirus may contribute to carcinogenesis [111].

Several mechanisms of possible involvement of HCMV with tumor transformation have been proposed. HCMV possesses several viral proteins interfering with cellular processes and, thus, can stimulate cell migration, proliferation, and inhibition of apoptosis [105]. Available evidence suggests that HCMV may be involved in modulating the microenvironment of the head and neck tumors through its action on tumor cells [111].

### 4.4. Role of Herpes Simplex in Oral Cancer

There are two closely related forms of the herpes simplex virus (HSV) that are known as HSV-1 and HSV-2. The action of HSV-1 has a particular affinity to oral infections, while HSV-2 is rather related to genital infections [106].

Jalouli et al. detected HSV in 15% of oral carcinoma samples out of the total of 155 tumors [112]. Another study from 2010 statistically associated HSV infection with oral carcinogenesis, and the concentration of antibodies against HSV-1 and HSV-2 in patients with OSCC was elevated [113]. These findings may be in accordance with the recent report that HSV is able to act as a mutagen toward host cells [106].

## 5. Findings and Discussion

Hopper et al. reported the existence of viable bacteria in deep parts of OSCC; some species were isolated only from either tumorous or nontumorous tissue samples [13]. This implies that bacteria, especially facultatively anaerobic and anaerobic bacteria, may survive in the tumor microenvironment. In 2019, Al-Hebshi et al. suggested a “passenger-turning-driver” conceptual model of the sustainable role of the oral microbiota in oral cancer [114]. According to this model, the expression of proinflammatory microbial features and virulence factors creates a functionally dysbiotic “driver” intratumor microbiota that enhances the progression of oral cancer.

Here, we summarize the current knowledge about the relationships between individual microorganisms and OSCC. As shown in recent studies, many bacteria, fungi, and viruses can possibly play a role in OSCC etiopathogenesis. Association studies included in this review were designed as case–control or case–case studies, analyzing the bacteriome and mycobiome from saliva, oral rinses, oral mucosal swabs, or oral mucosal tissue samples (deep and superficial) and comparing the results in healthy individuals to those with OSCC and/or with premalignant lesions.

The bacteriome and mycobiome were analyzed using methods based predominantly on 16S or ITSs rRNA sequencing. However, a certain inconsistency in results across studies, which is a consequence of different sample handling and the differences in laboratory procedures or in statistical processing of the acquired datasets, still remains a limitation of sequencing analysis [115]. Therefore, it is necessary to standardize the sampling procedures by establishing guidelines supporting future comparisons of similar studies. Hence, to acquire objective results, there is an increasing need for further sequencing analyses and for meta-analyses standardizing large amounts of data. Such meta-analyses based on sequencing analysis could lead to the identification of a comprehensive microbial profile associated with oral cancer and, thus, to a better understanding of the role of the oral microbiota in the etiopathogenesis of oral cancer.

In the tumoral tissue and saliva from patients with OSCC, relative abundances of Gram-negative anaerobic bacteria, mostly *F. nucleatum* [43,46,47] and *P. gingivalis* [12,14,46], were increased compared to samples from healthy individuals. A similar association with OSCC was reported for aerobic Gram-negative bacteria *Campylobacter* sp. [41,43]. In premalignant oral lesions, the representation of *Fusobacterium* and *Campylobacter* spp. in comparison to the healthy tissues of the same patients was also elevated [116]. Generally, the increased relative abundances of periodontal pathogens (*P. gingivalis*, *F. nucleatum*, *P. micra* [42], and *Treponema* sp. [41,47]) in the oral cavity seem to be a risk factor for OSCC development. Gallimindi et al. have even demonstrated in a mouse model with chemically induced OSCC that *P. gingivalis* and *F. nucleatum* can promote oral carcinogenesis [117].

Decreased relative abundance of specific Gram-positive Actinomycetales bacteria [40,41,42,48], comprising aerobic/facultatively anaerobic genera of *Actinomyces*, *Rothia*, and *Corynebacteria*, was observed in samples of oral mucosal swabs and oral rinses of OSCC patients compared to controls. In the case of facultatively anaerobic Gram-positive bacterium *S. mitis*, controversial associations were reported, depending on the type of the analyzed sample (oral rinse, oral mucosal tissue, and saliva) [14,42,43]. From this perspective, a comparison of microbiomes from patients with OSCC (multiple sites in each patient, such as healthy mucosa, a swab from the surface of the tumor, internal tumor sample) could yield highly interesting results.

This review also summarizes the current knowledge about oral fungi and viruses. In our opinion, it is important to analyze the full microbiome, not only the bacteriome. *Candida* spp. were associated with OSCC; increased relative abundances of this fungi were observed in saliva samples from patients with OSCC compared to those taken from healthy individuals and, similarly, in tumoral tissue compared to fibroepithelial polyps [82,83]. A microbial coexistence of *C. albicans* and a periodontal pathogen *P. gingivalis* has been described; these two microorganisms have a relationship in which *C. albicans* shields the *P. gingivalis* from recognition by the host immune system and, in this way, supports the bacterial infection [118]. Moreover, *C. albicans* displays a synergistic interaction with commensal oral streptococci, which can have implications for the pathogenic potential of *C. albicans* in the oral mucosa [119].

The relationship between the prevalence of oral human papillomavirus DNA and periodontal disease is known [120]; in particular, *P. gingivalis* and *F. nucleatum* and HPV interact and, as a result of certain inflammatory reactions triggered by these microorganisms, this association further leads to the initiation and progression of head and neck squamous cell carcinoma and potentially OSCC. In vitro studies revealed that HPV oncogene expression (E6/E7) increases when virally infected cells are exposed to metabolites from these periodontal pathogens. Furthermore, increased histone modifications surrounding the viral promoter that are associated with an increase in viral transcripts and an overall open chromatin conformation were found in the presence of *P. gingivalis* and *F. nucleatum* [121]. Núñez-Acurio et al. also suggested a bidirectional interaction between EBV-positive oral epithelial cells and *P. gingivalis*, and they proposed the mechanisms via which these microorganisms can act independently or cooperate synergistically in the development of OSCC [122].

There are a wide range of mechanisms through which microorganisms can interact with components of the host cellular regulatory systems. The oral microbiota has been linked to tumor formation through the stimulation of cell proliferation, tumor invasiveness, and tumor angiogenesis, as well as through the inhibition of cell apoptosis, induction of chronic inflammation, or production of oncometabolites. These mechanisms of microbial action are presented in Figure 1.

The relative abundance of specific microorganisms may be directly controlled by the nutritional state of the patient. The human immune system has coevolved with the human microbiota, in a way “managing” these microorganisms [123] and using them as the source of metabolites [124]. The immune system ascertains that most infections caused by viruses or bacteria are self-limiting [125,126]. Yet, as the human immune system also contributes to the acquisition of nutrition from the human microbiota, overnutrition may occur if one eats normally without restriction [127,128]. This overnutrition may cause lipotoxicity and tissue damage [129,130], which may promote chronic inflammation, fuel microbial dysbiosis (excessive growth of certain microorganisms), and lead to chronic diseases including cancer [128,131].

Crosstalk between the tumor and the microenvironment has been intensively studied, with the recognition of the fact that complex interactions exist among tumor cells, stromal cells, overall host immunity, host microbiota, and the environment surrounding the host. Each of these variables is important as they relate to the treatment response, as is the view of their complex actions and interactions [132]. Recently, Li et al. suggested that some oral bacteria exert effects on chemotherapeutic drugs and influence the potential curative effect [16]. For example, it was described that oral cancer cells sustainedly infected with *P. gingivalis* exhibit resistance to paclitaxel [133] and have higher metastatic potential. Thus, a deeper understanding of the influence of the tumor microbiome on the behavior and response of tumors to systemic therapy may help improve its clinical efficacy [134]. Because potential stage-specific bacteria were identified (for example, *Neisseria elongata*, the abundance of which increased exclusively in OSCC stage 4) [42], alterations in the composition of the oral microbiome are especially considered as potential diagnostic and prognostic biomarkers for oral cancer [9].

## 6. Conclusions

This review presented an overview of the associations between individual oral bacteria, fungi, and viruses and OSCC. The novelty of this review lies in the complex approach to the oral microbiota as a whole, i.e., the inclusion of studies investigating several types of oral microorganisms, and the possible mechanisms of their contributions to the OSCC development and progression. Furthermore, the mechanisms via which specific oral microorganisms may potentially contribute to chronic inflammation, malignant transformation, and the development of oral cancer were outlined. Understanding the role of the complex oral microbiota in the etiopathogenesis of OSCC, as well as the associated molecular and cellular mechanisms, may be crucial for detection of the disease progression or relapse and possibly for the effectiveness of treatment.

## Figures and Tables

**Figure 1 microorganisms-09-01549-f001:**
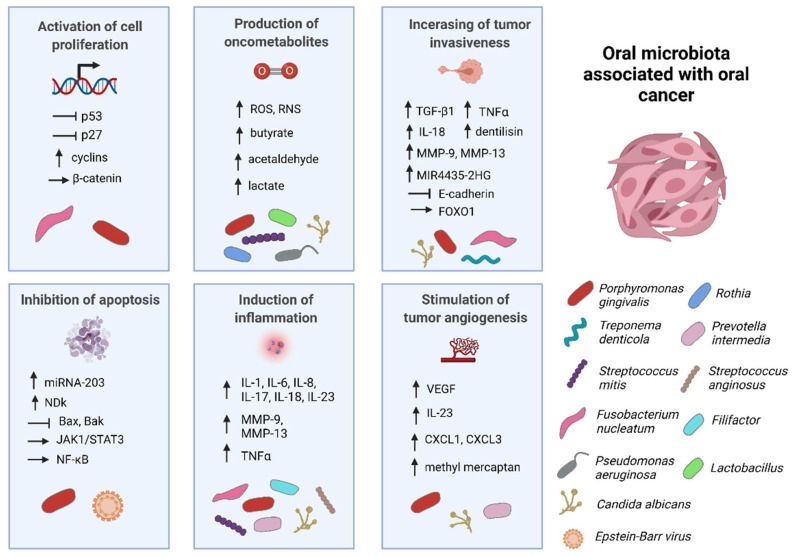
Graphical representation of the mechanisms via which selected microorganisms may be associated with oral squamous cell carcinoma (OSCC). The described mechanisms are discussed in detail in this review. Abbreviations: ROS, reactive oxygen species; RNS, reactive nitrogen species; TGF-β1, transforming growth factor beta 1; TNFα, tumor necrosis factor α; IL, interleukin; MMP-9, matrix metalloproteinase-9; MMP-13, matrix metalloproteinase-13; FOXO1, forkhead box O1; NDk, nucleoside diphosphate kinase; JAK1/STAT3, janus kinase 1/signal transducer and activator of transcription 3; VEGF, vascular endothelial growth factor; CXCL1, C–X–C motif chemokine 1; CXCL3, C–X–C motif chemokine 3; NF-κB, nuclear factor kappa B.

**Table 1 microorganisms-09-01549-t001:** Possible mechanisms of association between oral bacteria and oral cancer: in vitro studies performed on human samples.

Genus/Species	Possible Mechanisms of Association with Oral Cancer
*Acetobacter syzygii*	Possesses anticancer activity promoting the induction of apoptosis in oral cancer cells [22]
*Actinobacillus*	Upregulation of CCL20 in cancer cells [23]
*Aggregatibacter*	Production of proinflammatory cytokines [4]; production of hydrogen sulfide and methyl mercaptan inducing inflammation, cell proliferation, and tumor angiogenesis [9]
*Capnocytophaga*	Stimulation of inflammation [4]
*Catonella*	Induction of chronic inflammation [15]
*Eikenella corrodens*	Elevated production of IL-1, IL-6, IL-8, and TNF-α [24]
*Enterococcus*	Increase in genomic instability linked to superoxide production [25]; maintaining chronic inflammation [26]
*Filifactor*	Production of proinflammatory cytokines; activation of oncogenes; enhances tumor progression by promoting colonization by other pathogens [9]
*Fusobacterium nucleatum*	Secretion of IL-1β through activation of NLRP3 inflammasome [27]; p38 activation leading to increased production of MMP-13 and MMP-9 [28]; Ku70/p53 signaling pathway-dependent DNA damage. [29]; acceleration of cell cycle through p27 downregulation; induction of epithelial–mesenchymal transition through lncRNA MIR4435-2HG/miR-296-5p/Akt2/SNAI1 pathway [30]; activation of oncogenes cyclin D1 and myc through β-catenin pathway [31]
*Gemella*	IL-23 upregulation [17]
*Lactobacillus*	Some species produce lactate; *L. fermentum* produces hydrogen peroxide [32];
	*L. planarum* induces cancer cell apoptosis via upregulation of PTEN and downregulation of MAPK pathway [16]
*Mycoplasma salivarium*	p53 inhibition; activation of NF-κB signal pathway [33]
*Parvimonas*	Inflammation induction [16]
*Porphyromonas gingivalis*	Stimulation of Jak1/Stat3 signaling pathway through upregulation of proinflammatory cytokines [27]; upregulation of miRNA-203 [34]; production of nucleoside diphosphate kinases [35]; stimulation of cell proliferation through upregulation of cyclins and p53 inhibition [28]; induction of epithelial–mesenchymal transition through overexpression of β-catenin; chronic inflammation induction through IL-8, IL-6, TGF-β1, and TNF-α expression [36]; production of reactive oxygen species, butyrate, and acetaldehyde [37]
*Prevotella intermedia*	Production of virulent factors (lipopolysaccharides, peptidoglycans, lipoteichoic acid [9]; IL-1, IL-6, IL-17, IL-23, and TNF-α expression [22]; secretion of proteases [15]; production of hydrogen sulfide, methyl mercaptan, and acetaldehyde [4]
*Propionibacterium*	Production of IL-6 and IL-8 [4]
*Pseudomonas aeruginosa*	Induction of inflammation through NF-κB pathway activation [9]; DNA break induction leading to chromosomal instability; secretion of LasI factor leading to downregulation of E-cadherin expression [9]; endotoxins such as LPS or flagella contribute to the induction of inflammation [16]
*Rothia*	Acetaldehyde production [9]
*Streptococcus anginosus*	Production of proinflammatory cytokines; nitric oxide and cyclooxygenase-2 production [9]; acetaldehyde production [38]
*Streptococcus aureus*	Upregulation of COX-2 transcription; production of prostaglandins PGE2; induction of cyclin D1 overexpression [16]
*Streptococcus gordonii*	Suppression of epithelial–mesenchymal transition through decreasing ZEB2 expression; acetaldehyde production [36]
*Streptococcus mitis*	Suppression of OSCC cell proliferation in vitro [37]; prevents colonization by virulent microorganisms [4]; acetaldehyde production [38]
*Streptococcus salivarius*	Acetaldehyde production [9]
*Streptomyces*	Induction of cancer cell apoptosis [21]
*Tannerella*	Proinflammatory cytokine production [4]
*Treponema denticola*	Dentilisin overexpression associated with increased tumor invasiveness [18]

Abbreviation: CCL20, C–C motif chemokine ligand 20; NLRP3; NLR family pyrin domain-containing 3; lncRNA, long noncoding RNA; PTEN, phosphatase and tensin homolog; MAPK, mitogen-activated protein kinase; NF-κB; nuclear factor kappa B; LPS, lipopolysaccharide; COX-2, cyclooxygenase-2; PGE2, prostaglandin E_2_; ZEB2, zinc finger E-box = binding homeobox 2; TGF-β1, transforming growth factor beta 1; TNFα, tumor necrosis factor α; IL, interleukin; MMP-9, matrix metalloproteinase-9; MMP-13, matrix metalloproteinase-13; JAK1/STAT3, janus kinase 1/signal transducer and activator of transcription 3.

**Table 2 microorganisms-09-01549-t002:** Summary of mycobiota occurring in patients with oral cancer. Comparison of specific fungi abundance between case and control/case samples. All specimens were human.

Genus/Species	Abundance in OSCC Case Samples Relative to Control/Case Samples	Case Samples from OSCC Patients	Control/Case Samples	Number of Participants	Reference
*Aspergillus tamarii*	Decreased	OSCC tissue	FEP	25 OSCC patients, 27 FEP patients	[82]
*Alternaria*	Decreased	OSCC tissue	FEP	25 OSCC patients, 27 FEP patients	[82]
*Candida albicans*	Increased	OSCC tissue	FEP	25 OSCC patients, 27 FEP patients	[82]
*Candida etchellsii*	Increased	OSCC tissue	FEP	25 OSCC patients, 27 FEP patients	[82]
*Candida famata*	Increased	Saliva	Saliva	97 OSCC patients, 200 OPMD patients, 200 healthy individuals	[83]
*Cladosporium* *halotolerans*	Decreased	OSCC tissue	FEP	25 OSCC patients, 27 FEP patients	[82]
*Emericella*	Decreased	Tongue cancer tissue	Normal tissue	39 OSCC patients	[84]
*Gibberella*	Increased	OSCC tissue	FEP	25 OSCC patients, 27 FEP patients	[82]
*Hannaella*	Increased	OSCC tissue	FEP	25 OSCC patients, 27 FEP patients	[82]
*Malassezia restricta*	Decreased	OSCC tissue	FEP	25 OSCC patients, 27 FEP patients	[82]
*Pichia anomala*	Decreased	Saliva	Saliva	97 OSCC patients, 200 OPMD patients, 200 healthy individuals	[83]
*Rhodotorula* *mucilaginosa*	Increased	OSCC tissue	FEP	25 OSCC patients, 27 FEP patients	[82]
*Trametes*	Decreased	OSCC tissue	FEP	25 OSCC patients, 27 FEP patients	[82]

Abbreviations: FEP, fibroepithelial polyps; OSCC, oral squamous cell carcinoma; OPMD, oral potentially malignant disorders.

**Table 3 microorganisms-09-01549-t003:** An overview of viruses detected in samples from patients with oral cancer. All specimens were human.

Virus Type	Samples from OSCC Patients	Conclusion	Number of Participants ^1^	Reference
*HPV*	Saliva and OSCC tissue	HPV 16 positivity rate was 15.4% (saliva), HPV 18 positivity rate was 1.6% (tissue)	135 samples (13 saliva, 59 blood, 63 OSCC tissues)	[97]
	Paraffin-embedded OSCC tissue	HPV 16 positivity rate was 19.2%	114 OSCC patients	[98]
	Saliva	46% of patients had positive HPV-DNA	35 OSCC patients, 20 healthy individuals	[99]
	FFPE OSCC tissue	66% of cases were positive for HPV 38 DNA	53 OSCC patients	[100]
	Samples of OSCC, oral leukoplakia, oral lichen planus	The frequency of HPV positivity was 1.54%	32 OSCC patients, 17 patients with oral leukoplakia, 16 patients with oral lichen planus	[101]
*EBV*	FFPE OSCC tissue	The prevalence of EBV in OSCC was 41.2%	165 OSCC patients	[102]
	OSCC tissue	Microarray analysis found 82.5% EBV prevalent rate	57 OSCC patients	[103]
	OSCC tissue	20% of cases were positive for EBV	20 OSCC patients, 20 controls	[104]
*HCMV*	FFPE OSCC tissue	6.3% of cases were positive for HCMV	48 OSCC patients	[105]
*HSV*	FFPE OSCC tissue	HSV-1 was detected in 22% of cases, HSV-2 in 8% of cases	40 OSCC patients, 10 patients with benign tumor	[106]

^1^ If the number of patients was not known, the number of samples was added. Abbreviations: OSCC, oral squamous cell carcinoma; FFPE, formalin-fixed paraffin-embedded; HPV, human papillomavirus; EBV, Epstein–Barr virus; HCMV, human cytomegalovirus; HSV, herpes simplex virus.

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
