# Peer review of "The Role of the Oral Microbiota in the Etiopathogenesis of Oral Squamous Cell Carcinoma"

_microorganisms, 2021, doi:10.3390/microorganisms9081549_

Round 1

Reviewer 1 Report

This review article presents an overview on the associations between individual oral microbiota like bacteria, fungi, and viruses with OSCC. The authors suggest that changes in relative abundances of specific bacteria, viruses and fungi may be associated with OSCC, and oral microorganisms may contribute to chronic inflammation, malignant transformation, and the development of oral cancer.

The following points may be taken into consideration when the authors revise the manuscript. The relative abundance of specific microorganisms may be directly controlled by the nutritional state of the patient. The human immune system has co-evolved with the human microbiota in managing these microorganisms [1] and using them as source of metabolites [2]. Because of this immune system, most of the infections caused by viruses or bacteria are self-limiting [3,4]. Yet, as the human immune system also contributes to nutrition acquisition from the human microbiota, over-nutrition may occur if one eats normally without restriction [5,6]. This over-nutrition may cause lipotoxicity and tissue damage [7,8], which may promote chronic inflammation and fuel microbial dysbiosis (over-growth of specific microorganisms) and lead to chronic diseases like cancer [6,9]. So the interplay of nutritional state of the patients with microbiota and cancer pathogenesis should also be investigated and reviewed.

Careless mistake in the manuscript:

  1. Page 2, line 80, “detected OSCC tissue samples” should be “in OSCC tissue samples”.

The following references may be included in the revised manuscript to provide a better view on the issue:

  1. Broderick, N.A. A common origin for immunity and digestion. Front. Immunol. 2015, 6, 72. [CrossRef]
  2. McFall-Ngai, M.; Hadfield, M.G.; Bosch, T.C.; Carey, H.V.; Domazet-Loso, T.; Douglas, A.E.; Dubilier, N.; Eberl, G.; Fukami, T.; Gilbert, S.F.; et al. Animals in a bacterial world, a new imperative for the life sciences. Proc. Natl. Acad Sci. USA 2013, 110, 3229–3236.
  3. Levin, B.R.; Antia, R. Why we don’t get sick: The within-host population dynamics of bacterial infections. Science 2001, 292, 1112–1115.
  4. Levin, B.R.; Baquero, F.; Ankomah, P.P.; McCall, I.C. Phagocytes, Antibiotics, and Self-Limiting Bacterial Infections. Trends Microbiol. 2017, 25, 878–892.
  5. Troisi, J.; Venutolo, G.; Pujolassos Tanya, M.; Delli Carri, M.; Landolfi, A.; Fasano, A. COVID-19 and the gastrointestinal tract: Source of infection or merely a target of the inflammatory process following SARS-CoV-2 infection? World J. Gastroenterol. 2021, 27, 1406–1418.
  6. Howell MC, Green R, McGill AR, Dutta R, Mohapatra S, Mohapatra SS. SARS-CoV-2-Induced Gut Microbiome Dysbiosis: Implications for Colorectal Cancer. Cancers. 2021; 13(11):2676.
  7. Saltiel, A.R.; Olefsky, J.M. Inflammatory mechanisms linking obesity and metabolic disease. J. Clin. Investig. 2017, 127, 1–4.
  8. Garbarino, J.; Sturley, S.L. Saturated with fat: New perspectives on lipotoxicity. Curr. Opin Clin. Nutr. Metab. Care 2009, 12, 110–116.
  9. Whisner, C.M.; Athena Aktipis, C. The Role of the Microbiome in Cancer Initiation and Progression: How Microbes and Cancer Cells Utilize Excess Energy and Promote One Another’s Growth. Curr. Nutr. Rep. 2019, 8, 42–51.

Author Response

Specific changes made in manuscript ID: microorganisms-1302526 - revision

To: Reviewer 1

Comment 1: This review article presents an overview on the associations between individual oral microbiota like bacteria, fungi, and viruses with OSCC. The authors suggest that changes in relative abundances of specific bacteria, viruses and fungi may be associated with OSCC, and oral microorganisms may contribute to chronic inflammation, malignant transformation, and the development of oral cancer.

The following points may be taken into consideration when the authors revise the manuscript. The relative abundance of specific microorganisms may be directly controlled by the nutritional state of the patient. The human immune system has co-evolved with the human microbiota in managing these microorganisms [1] and using them as source of metabolites [2]. Because of this immune system, most of the infections caused by viruses or bacteria are self-limiting [3,4]. Yet, as the human immune system also contributes to nutrition acquisition from the human microbiota, over-nutrition may occur if one eats normally without restriction [5,6]. This over-nutrition may cause lipotoxicity and tissue damage [7,8], which may promote chronic inflammation and fuel microbial dysbiosis (over-growth of specific microorganisms) and lead to chronic diseases like cancer [6,9]. So the interplay of nutritional state of the patients with microbiota and cancer pathogenesis should also be investigated and reviewed.

The following references may be included in the revised manuscript to provide a better view on the issue:

  1. Broderick, N.A. A common origin for immunity and digestion. Front. Immunol. 2015, 6, 72. [CrossRef]

  2. McFall-Ngai, M.; Hadfield, M.G.; Bosch, T.C.; Carey, H.V.; Domazet-Loso, T.; Douglas, A.E.; Dubilier, N.; Eberl, G.; Fukami, T.; Gilbert, S.F.; et al. Animals in a bacterial world, a new imperative for the life sciences. Proc. Natl. Acad Sci. USA 2013, 110, 3229–3236.

  3. Levin, B.R.; Antia, R. Why we don’t get sick: The within-host population dynamics of bacterial infections. Science 2001, 292, 1112–1115.

  4. Levin, B.R.; Baquero, F.; Ankomah, P.P.; McCall, I.C. Phagocytes, Antibiotics, and Self-Limiting Bacterial Infections. Trends Microbiol. 2017, 25, 878–892.

  5. Troisi, J.; Venutolo, G.; Pujolassos Tanya, M.; Delli Carri, M.; Landolfi, A.; Fasano, A. COVID-19 and the gastrointestinal tract: Source of infection or merely a target of the inflammatory process following SARS-CoV-2 infection? World J. Gastroenterol. 2021, 27, 1406–1418.

  6. Howell MC, Green R, McGill AR, Dutta R, Mohapatra S, Mohapatra SS. SARS-CoV-2-Induced Gut Microbiome Dysbiosis: Implications for Colorectal Cancer. Cancers. 2021; 13(11):2676.

  7. Saltiel, A.R.; Olefsky, J.M. Inflammatory mechanisms linking obesity and metabolic disease. J. Clin. Investig. 2017, 127, 1–4.

  8. Garbarino, J.; Sturley, S.L. Saturated with fat: New perspectives on lipotoxicity. Curr. Opin Clin. Nutr. Metab. Care 2009, 12, 110–116.

  9. Whisner, C.M.; Athena Aktipis, C. The Role of the Microbiome in Cancer Initiation and Progression: How Microbes and Cancer Cells Utilize Excess Energy and Promote One Another’s Growth. Curr. Nutr. Rep. 2019, 8, 42–51.

Answer 1: Thank you very much for your comment, the text you suggested was, with some minor alterations, included in the discussion of our manuscript (page 27, lines 662-671).

Comment 2: Careless mistake in the manuscript: Page 2, line 80, “detected OSCC tissue samples” should be “in OSCC tissue samples”.

Answer 2: Thank you, the preposition was aded into the text (page 2, line 71).

Comment 3: English language and style are fine/minor spell check required.

Answer 3: The English of the manuscript was further improved by a professional agency.

Thank you very much for your comments.

Reviewer 2 Report

Summary:

The article is focused on analyzing the role of oral microorganisms in the development of oral squamous cell carcinoma, a condition with a high incidence nowadays.  In general terms, this is not a genuine topic since several similar reviews have been published already quite recently [1-3]. However, there are some aspects that differentiate this article from other previous publications. Specifically, it gives a broader vision of the main underlying mechanisms described, that may be triggered by the different types of oral microorganisms (including bacteria, fungi and virus). Although a few aspects are susceptible to being improved, this literature review is reasonably organized on its layout and clear, besides, it demonstrates an adequate justification of the displayed evidence.

Major criticisms

  • Some sections of the articles are written in imprecise English that may be easily improved to facilitate its comprehension and understanding (particularly from line 78 to 125).
  • On the other hand, in the abstract and in the discussion section, it is stated that “association studies included in this review were designed as case-control/case studies” and that is not completely true. In the article there are many references to literature reviews that are not case-control or case studies.

Minor criticisms

  • As mentioned above, the topic of this review is not characterized by its novelty. In fact, some of the literature included and cited show very similar information.
  • The discussion section includes 4 paragraphs (from line 604 to 647), that describe information that may be considered rather “results” and will suit better in the previous sections.
  • The abstract should not state a personal opinion and therefore I would suggest deleting the “in our opinion” sentence written in line 29.

Article strengths

Despite the criticisms, it is worth mentioning that there is a noticeable work and effort on this manuscript. The information provided is reasonably complete, significantly worked out and detailed. Moreover, the paper is based on an extensive contracted bibliography.

Similarly, I would like to congratulate the authors for the job done on designing the graphical abstract as it gives a simple, but very visual and schematic, view of the major findings noted in this review.

Other suggestions

  • The article might benefit from a shorter title to better capture the readers’ attention. For this, the use of the terms “oral microbiome” or “oral microbiota” could replace “oral bacteria, fungi and viruses”.

There are other points that should be stated in order to help the achievement of a better article quality:

  • Ideally, a literature review should not include other literature reviews when access to the original research is available. It is not mandatory but recommended.
  • Table 1 would be a lot more explicit and useful if it stated whether the results obtained were originated from in vitro or in vivo experimenting, or, conversely, from the analysis of human samples. The rationale for this suggestion lies in the fact that there are a number of limitations described on in vitro experimenting at a molecular level, but also with respect to the oral microbiome in experimental animals, it is possible to assure that it differs substantially from the oral microbiome in humans. Therefore, the handling of information on oral microbiota obtained from animal experimentation should always be done considering the limitations to extrapolate results to humans.

[1] Sami A, Elimairi I, Stanton C, Ross RP, Ryan CA. The Role of the Microbiome in Oral Squamous Cell Carcinoma with Insight into the Microbiome-Treatment Axis. Int J Mol Sci. 2020;21(21):8061. Published 2020 Oct 29. doi:10.3390/ijms21218061

[2] Li Q, Hu Y, Zhou X, Liu S, Han Q, Cheng L. Role of Oral Bacteria in the Development of Oral Squamous Cell Carcinoma. Cancers (Basel). 2020;12(10):2797. Published 2020 Sep 29. doi:10.3390/cancers12102797

[3] Gholizadeh P, Eslami H, Yousefi M, Asgharzadeh M, Aghazadeh M, Kafil HS. Role of oral microbiome on oral cancers, a review. Biomed Pharmacother. 2016 Dec;84:552-558. doi: 10.1016/j.biopha.2016.09.082. Epub 2016 Sep 29. PMID: 27693964

Author Response

Specific changes made in manuscript ID: microorganisms-1302526 - revision

To: Reviewer 2

Comment 1: The article is focused on analyzing the role of oral microorganisms in the development of oral squamous cell carcinoma, a condition with a high incidence nowadays.  In general terms, this is not a genuine topic since several similar reviews have been published already quite recently [1-3]. However, there are some aspects that differentiate this article from other previous publications. Specifically, it gives a broader vision of the main underlying mechanisms described, that may be triggered by the different types of oral microorganisms (including bacteria, fungi and virus). Although a few aspects are susceptible to being improved, this literature review is reasonably organized on its layout and clear, besides, it demonstrates an adequate justification of the displayed evidence.

Despite the criticisms, it is worth mentioning that there is a noticeable work and effort on this manuscript. The information provided is reasonably complete, significantly worked out and detailed. Moreover, the paper is based on an extensive contracted bibliography.

Similarly, I would like to congratulate the authors for the job done on designing the graphical abstract as it gives a simple, but very visual and schematic, view of the major findings noted in this review.

Answer 1: Thank you very much for your positive evaluation of our work.

Major criticisms

Comment 2: Some sections of the articles are written in imprecise English that may be easily improved to facilitate its comprehension and understanding (particularly from line 78 to 125).

Moderate English changes required.

Answer 2: Based on this comment, the manuscript was sent to a professional agency for proofreading, see pages 2-3, lines 68-114 .

Comment 3: On the other hand, in the abstract and in the discussion section, it is stated that “association studies included in this review were designed as case-control/case studies” and that is not completely true. In the article there are many references to literature reviews that are not case-control or case studies.

Answer 3: Well, we did not claim that the review contains solely case-control/case studies. We stated that the association studies that have been included (among other types of papers) in the manuscript are case-control/case studies. Numbers of patients/samples were added to Tables S1, 2, and 3.

Minor criticisms

Comment 4: As mentioned above, the topic of this review is not characterized by its novelty. In fact, some of the literature included and cited show very similar information.

Answer 4: Unlike other reviews focusing on microbiota in the oral cavity, our paper does not focus just on certain types of microorganisms but it is a comprehensive review that describes associations between OSCC and bacteria, fungi and viruses. Moreover, the review focuses on mechanisms that may cause he involvement of the respective microorganisms in etiopathogenesis (summarized in Figure 1). Based on this criticism, we attempted to highlight the novelty of this paper also in the conclusion (page 17, lines 688-691).

Comment 5: The discussion section includes 4 paragraphs (from line 604 to 647), that describe information that may be considered rather “results” and will suit better in the previous sections.

Answer 5: The chapter was renamed to “Findings and Discussion”.

Comment 6: The abstract should not state a personal opinion and therefore I would suggest deleting the “in our opinion” sentence written in line 29.

Answer 6: The sentence in the abstract was corrected (page 1, lines 18-20).

Other suggestions

Comment 7: The article might benefit from a shorter title to better capture the readers’ attention. For this, the use of the terms “oral microbiome” or “oral microbiota” could replace “oral bacteria, fungi and viruses”.

 Answer 7: Thank you for the suggestion, we have actually given this issue much thought even before submission. In the end, we have decided on the full title mentioning bacteria, funghi, and viruses as the word “microbiota” is more commonly associated (although incorrectly) with bacteria only. Nevertheless, we also agree that the title became too long and, hence, the title of the review was changed to “The role of the oral microbiota in the etiopathogenesis of oral squamous cell carcinoma”.

There are other points that should be stated in order to help the achievement of a better article quality:

Comment 8: Ideally, a literature review should not include other literature reviews when access to the original research is available. It is not mandatory but recommended.

 Answer 8: This is partly true; nevertheless, we have used these reviews for cross-referencing and they became a valuable source of information for our paper. For this reason, we believe that it would be unfair to capitalize on the work done by their authors and not even to cite them.

Comment 9: Table 1 would be a lot more explicit and useful if it stated whether the results obtained were originated from in vitro or in vivo experimenting, or, conversely, from the analysis of human samples. The rationale for this suggestion lies in the fact that there are a number of limitations described on in vitro experimenting at a molecular level, but also with respect to the oral microbiome in experimental animals, it is possible to assure that it differs substantially from the oral microbiome in humans. Therefore, the handling of information on oral microbiota obtained from animal experimentation should always be done considering the limitations to extrapolate results to humans. 

Answer 9: We agree with the reviewer; the following information was added into Table 1 “in vitro studies performed on human samples” (page 3, lines 116-117).

Thank you very much for your comments.

Round 2

Reviewer 2 Report

I highly appreciate that you have considered the advices given in the previous review and that you have clarified some of the aspects noted.

Currently the manuscript has improved its quality and refinement. However, a few (minimal) grammar errors have been found, specially on the highlighted sections that are typed below the "findings and discussion" subheading.